



# Water table driven greenhouse gas emission estimate guides peatland restoration at national scale

Julian Koch[1], Lars Elsgaard[2], Mogens H. Greve[2], Steen Gyldenkærne[3], Cecilie Hermansen[2], Gregor Levin[3], Shubiao Wu[2] & Simon Stisen[1]

[1]Geological Survey of Denmark and Greenland, Department of Hydrology, Copenhagen, Denmark

[2]Aarhus University, Department of Agroecology, Tjele, Denmark

[3]Aarhus University, Department of Environmental Science, Roskilde, Denmark

*Correspondence to*: Julian Koch (juko@geus.dk)

**Abstract.** The substantial climate change mitigation potential of restoring peatlands, through rewetting and extensifying agriculture to reduce greenhouse gas (GHG) emissions is largely recognized. The green deal in Denmark aims at restoring 100,000 ha peatlands by 2030. This area corresponds to more than half of the Danish peatland, with an expected reduction of GHG emissions of almost half of the entire land use, land use change and forestry (LULUFC) emissions. Recent advances established the functional relationship between hydrological regimes, i.e. water table depth (WTD), and $CO_2$ and $CH_4$ emissions. This builds the basis for science-based tools to evaluate and prioritize peatland restoration projects. With this article, we lay the foundation of such a development by developing a high-resolution WTD map for Danish peatlands. Further we define WTD repose functions ($CO_2$ and $CH_4$) fitted to Danish flux data to derive a national GHG emission estimate for peat soils. We estimate the annual GHG emissions to be 2.6 Mt CO2-eq, which is around 15% lower than previous estimates. Lastly, we investigate alternative restoration scenarios and identify substantial differences in the GHG reduction potential depending on the prioritization of fields in the rewetting strategy. If wet fields are prioritized, which is not unlikely in a context of a voluntary bottom-up approach, the GHG reduction potential is just 30% for the first 10,000 ha with respect to a scenario that prioritizes drained fields. This underpins the importance of the proposed framework linking WTD and GHG emissions to guide a spatially differentiated peatland restoration.

## 1 Introduction

The natural environmental conditions of peatlands represent a waterlogged, anoxic and often acidic soil ecosystem that favours the accumulation of organic carbon (C) due to impeded microbial mineralization of plant biomass. During the last centuries, anthropogenic induced changes of the environmental conditions have deteriorated the natural functioning of many peatlands



across the globe and have transformed them from an atmospheric carbon sink to a carbon source (Huang et al., 2021; Tiemeyer
et al., 2016; Wilson et al., 2015). Thus, in order to expand arable land, water tables were lowered, soils were limed and
inundation was prevented through establishment of artificial drainage and stream management. This has enhanced microbial
mineralization and $CO_2$ emissions. In consequence, drained peatlands are accountable for approx. 1 Gt $CO_2$ equivalents ($CO_2$-eq) per year at global scale, which corresponds to 10% of the total greenhouse gas (GHG) emissions from the land use, land
use change and forestry (LULUFC) sectors (Smith et al., 2014). It is widely acknowledged that targeted management of
peatlands is needed to mitigate their contribution to climate change (Hambäck et al., 2023; Wilson et al., 2016).

The emissions of $CO_2$ and methane ($CH_4$) are linked to hydrologic regimes where a deeper water table favours $CO_2$ emissions
and a very shallow water table permits $CH_4$ emissions (Evans et al., 2021; Tiemeyer et al., 2020). The functional relationship
between nitrous oxide ($N_2O$) and water table depth is less certain (Tiemeyer et al., 2020), however full saturation is typically
linked to zero or negligible $N_2O$ emissions from organic soils, such as <1 kg $N_2O$ ha$^{-1}$ year$^{-1}$ (Minkkinen et al., 2020; Wilson
et al., 2016). It is widely recognised that restoring cultivated peatlands by rewetting is a robust climate mitigation strategy,
although the ecosystems may not reach their natural environmental conditions on short term (Audet et al., 2013; Kandel et al.,
2018). In practice, such restoration implies at a minimum to cease tillage and to reduce artificial drainage, e.g., by deregulating
streams, blocking of drain pipes and ditches.

To mitigate agricultural GHG emissions, and to improve nature quality and biodiversity, Danish ministerial agreements were
launched in 2021 to restore 100,000 ha peatland by 2030. It has been estimated that the total of Danish peatlands (173,000 ha)
emit approx. 5.4 Mt $CO_2$-eq yr$^{-1}$, which is by far the largest source in the LULUCF sector (Nielsen et al., 2022). Further, it has
been suggested that the emissions could be potentially reduced by 4.1 Mt $CO_2$-eq through restoration (Klimarådet, 2020). Yet,
mitigation effects of large-scale peatland restoration remain uncertain, since precise knowledge of the baseline emissions is
missing, and tools are critically needed to guide the restoration by prioritising areas with the largest GHG reduction potential.
Oxygen status in the peat soil, as controlled by water saturation, is among the strongest proximal drivers of microbial
mineralization and losses of GHG (Karki et al., 2014). Therefore, large-scale models of water table depth (WTD) in peat soils
could potentially be a useful proxy for the intensity of GHG emissions, thereby contributing to guide national rewetting
initiatives (Tiemeyer et al., 2020).

There exists a suite of tools to model WTD in peat soils, namely process-based and conceptual models as well as data-driven
machine learning (ML) models. Modelling peatland WTD dynamics using process-based models requires site specific
knowledge on lateral flows of surface- and groundwater as well as correct representation of small-scale variability in
topography and soil properties (Bechtold et al., 2019; Gong et al., 2012). This poses challenges for large-scale modelling
applications. However, ML provides a suitable alternative, which can fully exploit available high-resolution geo-environmental
data sources and thereby bypassing the rigid parameterization and computational requirements of conventional hydrological
models. There exists a large body of literature addressing the applicability of ML to model site-specific temporal WTD
dynamics. However, to our knowledge, the potential of applying ML to model the spatial WTD variability at high-resolution
for large domains has only been investigated by few studies. Bechtold et al. (2014) applied boosted regression trees to model





a mean annual WTD for peatlands in Germany at a resolution of 25 m. Koch et al. (2019) modeled WTD for extreme winter conditions for a 15,000 km$^2$ domain in Denmark by using random forests. This work was later extended to national scale at 10

m resolution using gradient boosting decision trees for average summer and winter WTD (Koch et al., 2021). At smaller domains, Lendzioch et al. (Lendzioch et al., 2021) applied a random forest model to simulated WTD for two peat sites in the Czech Republic at sub-meter resolution using multi-spectral and thermal UAV data as input.

The present study is motivated by recent scientific advances in defining WTD response functions of $CO_2$ and $CH_4$ emissions and high-resolution ML based WTD modelling. The key objectives of the study are to (1) build a high-resolution ML based

WTD model for Danish peatlands, (2) define WTD response functions for $CO_2$ and $CH_4$ for Danish conditions, and (3) combine (1) and (2) to derive national scale GHG emission estimates and to showcase how the new knowledge can be used to support peatland restoration.

## 2 Data and methods

### 2.1 Study area

The study area covers the entire land area of Denmark, which corresponds to approx. 43,000 km$^2$. In order to restrict the domain for the data analysis and modelling to an area where WTD driven GHG upscaling may be of relevance, we calculated the union of two map layers that include a river valley bottom delineation (Sechu et al., 2021) and a map of wetlands (Greve et al., 2014). The two map layers correspond to approx. 775,000 ha and approx. 904,000 ha, respectively, and their union, which marks our model domain, amounts to approx. 1,162,000 ha, roughly one fourth of the total land area of Denmark (Figure

1). For the final analysis, the domain was further constrained to the carbon rich lowland soils. The total area of peat soils with organic content (OC) greater than 12% constitutes approx. 129,000 ha, of which approx. 74,000 are cultivated, either extensively (65%) or intensively (35%) (Greve et al., 2019; Levin, 2019).

### 2.2 Water table depth

A total of 24,492 WTD observations were assorted from various sources in order to compile a comprehensive training dataset

that reflects long-term average summer conditions. WTD observations recorded between the months of May and September in the period of 2000 to 2021 were used as training data. Figure 1 depicts the locations of the WTD observations and Table 1 provides an overview of the different sources and WTD statistics. Data from the Danish well database JUPITER *(Hansen and Pjetursson, 2011)* were processed by first constraining the well location to a 200 m buffer around the lowland soils. Second, only wells with a maximum filter depth of 5 m below ground were selected. The median WTD was used in case a well had

multiple observations within the specified period. This resulted in 5,716 WTD observations. Moreover, 4,796 WTD observations were obtained from two soil auger campaigns (2010 and 2021) that specifically targeted lowland soils with high organic content (Greve et al., 2014). 653 out of the 4,796 locations sampled in summer 2010 were revisited in summer 2021 and for the double sampled locations, the mean WTD was used in the final training dataset. The soil augering equipment was





limited to a maximum depth of 1.21 m and in case the water table was not detected by the auger, information could only be
derived for the minimum WTD at the given location. Further, 9,980 groundwater dependent lakes with a surface area greater
than 100 m² and located within a 200 m buffer around the peatland soils were used as proxy WTD observations. Since lake
water level observations were missing, values were drawn from a normal distribution with an assumed mean of -0.25 m, i.e.,
above terrain, and a standard deviation of 0.05 m. Additional dummy points for saturated conditions were placed along the
coastline and the river network. Here, 1,000 points for each category were placed randomly and assigned a WTD of 0 m.
Lastly, dummy points for drained conditions were generated along drain ditches and within drained forest. Here, 1000 points
for each category were placed randomly and sampled from a normal distribution with a mean WTD of 1.21 m and standard
deviation of 0.2 m. Figure 2 depicts the WTD variability of the training dataset differentiated for the data sources. The WTD
data derived from the national well database was the only source that contained deep WTD observations. The data originating
from the soil coring campaigns provided mostly shallow data; however, 3,110 samples had a WTD of 1.21 m and thereby
indicated solely a minimum WTD. The WTD of lakes is entirely above terrain, i.e., negative WTD values, whereas coast and
rivers were assigned a WTD of 0 m. We created a training dataset for summer conditions, because the WTD observations from
the soil auger campaigns are primarily from summer months and the WTD data from this source represent the primary
information on the shallow WTD, i.e., top meter below terrain. Based on the WTD observations from the national well
database, the median WTD for summer is 2.0 m whereas the median WTD for winter is 1.75 m, based on the same processing
as applied for the summer data, just for the months from October to March. The difference of 0.25 m can be understood as an
overall annual amplitude.

Based on a data synthesis by Tiemeyer et al. (2020), the WTD driven GHG response functions of $CO_2$ and $CH_4$ emissions
from organic soils exhibit a non-linear relationship with the most distinct sensitivity in the depth interval of 0 to 0.5 m. In
consequence, we aim at modelling WTD with the highest possible accuracy for this GHG sensitive WTD interval. With the
same motivation, Bechtold et al. (2014) presented a WTD transformation function that resulted in a pseudo linearity between
WTD and GHG. For our purpose, the transformation function presented by Bechtold et al. (2014) was adopted to:

$$WTD_t = \begin{cases} -1 * (e^{3*WTD} - 1) & WTD \geq 0 \\ -1 * (1 - e^{-3*WTD}) & WTD < 0 \end{cases}, \qquad \text{(Eq. 1)}$$

where $WTD_t$ is the transformed WTD. As shown in Figure 2, $WTD_t$ varies between -1 and 1, and reaches its upper asymptote
at a WTD of approximately 1 m. The applied WTD transformation also allows us to incorporate the 1.21 m WTD data, that
represent a minimum observation, since the $WTD_t$ variability above 1 m is minimal.

### 2.3 Covariates

A set of 27 covariates was curated to gather national scale map layers that are deemed relevant to explain the WTD variability
in the training dataset (Table 2). The individual maps were resampled from their native resolutions to the defined output
resolution of 10 m. The covariates encompassed high-resolution data on topography, water body proximity, lithology, land use
and hydrology. The water body proximity was expressed as both the vertical and horizontal distance to the nearest water body,



which contained rivers, lakes, and the coastline. Additionally, the vertical and horizontal distance to the nearest ditch was calculated to capture the effect of drainage on WTD in lowland soils. Using the historical crop type records, a six class ranked map indicating wetness of agricultural fields was created. The wetness rank represents a qualitative analysis based on agricultural expert judgement based on the Danish agricultural Land Parcel Information System for approx. 600,000 fields for

the years 2016 to 2020. Moreover, high-resolution data relevant to discriminate saturation conditions of the soil were obtained from Landsat and Sentinel-1 satellite systems, such as land surface temperature (LST), which serves as a valuable proxy for water-saturated soil conditions.

## 2.4 Machine learning model

We applied the CatBoost implementation of the well-established gradient boosting decision tree (GBDT) algorithm (Dorogush

et al., 2018; Prokhorenkova et al., 2018). In an additive training process, GBDT builds a prediction model based on an ensemble of weak learners, i.e., decision trees. For a pre-defined number of iterations, GBDT attempts to correct itself by adding a decision tree trained against the residuals of the ensemble sum of its predecessors. CatBoost is favorable over similar ML algorithms, such as Random Forests, Support Vector Machines, or other GBDT implementations (e.g., XGboost or LightGBM), with respect to computational time and memory usage, while achieving a competitive accuracy (Hancock &

Khoshgoftaar, 2020; Huang et al., 2019). The model is set up to predict WTD at a resolution of 10 m. Given the areal extent of the domain, over 116 million grid cells are simulated by the GBDT model. The cost function used in training the model was set to the root mean squared error (RMSE) and key hyper parameters were tuned via a randomized search. The following CatBoost hyper parameters were included in a simple randomized search with 2000 iterations: learning_rate, depth, subsample, rsm, l2_leaf_reg, min_data_in_leaf. The selected hyper parameters affect the overall architecture of individual trees as well as

limit the effect of overfitting. The best performing model, with respect to a 25% holdout validation was selected for subsequent final training. The final GBDT model was trained over 1000 iterations where 10% of the data were used as validation data to initiate early stopping once the validation cost function did not improve over 10 iterations. CatBoost allows assigning weights to the individual training data, which are used to calculate the cost function. In order to emphasize the GHG sensitive depth interval 0 to 0.5 m in the model training, a weight of 2 was assigned to shallow WTD observations.

The Shapley Additive exPlanations (SHAP) approach *(Lundberg and Lee, 2017)* was implemented to investigate the covariate importance of the trained GBDT model. SHAP builds upon game theory principles to explain the output of any ML model by quantifying marginal contributions of the applied covariates. SHAP values represent the contribution of each covariate to the final prediction and thereby provide valuable insights into trained ML models. The magnitude and sign of the SHAP values indicate the importance of a covariate and the direction of impact on the prediction, respectively. We calculated SHAP values

(i) for the training dataset to get insights into the trained GBDT model and (ii) for the prediction dataset to generate maps showing the relationships between covariates and WTD.





## 2.5 Synthesis and upscaling of Danish GHG flux data

The first measurements of $CO_2$ fluxes from cultivated peat soils in Denmark were performed in the 1970s using an in situ alkaline $CO_2$ trap method (Petersen et al., 1976), but it was not until 2008-2009 that a national monitoring campaign was
accomplished, where net fluxes of $CO_2$, $CH_4$ and $N_2O$ were measured at eight sites using state-of-the-art closed chamber techniques (Elsgaard et al., 2012; Petersen et al., 2012). Data on net ecosystem carbon balance from this campaign (Elsgaard et al., 2012) is used as the current Tier 2 emission factors (EFs) for organic soils with >12% OC in Denmark's National GHG Inventory report submitted under the United Nations Framework Convention on Climate Change and the Kyoto Protocol (Nielsen at al., 2021). National campaigns have not been repeated, but a number of research projects have generated additional
data on annual emissions of GHGs from Danish organic soils. A synthesis of these studies was performed in the present study (Supplementary Tables S1 and S2) and the data was used to derive response functions for GHG emissions in relation to WTD at mean annual conditions.

For analyzing $CO_2$ emissions, we employed a non-linear Gompertz function according to Tiemeyer et al. (2020):

$$CO_2\text{-}C(WTD) = CO_2\text{-}C_{min} + CO_2\text{-}C_{diff} * e^{-a*e^{b*WTD}} \qquad \text{(Eq. 2)}$$

where $CO_2\text{-}C_{min}$ is the lower asymptote, $CO_2\text{-}C_{diff}$ is the difference between upper and lower asymptote, a controls the displacement along the WTD axis and b defines the gradient. Indirect $CO_2$ emissions from leaching of dissolved organic carbon ($CO_2\text{-}C_{DOC}$) were added to Eq. 2, based on standard EFs of 0.31 Mg C ha$^{-1}$ for drained soils and 0.24 Mg C ha$^{-1}$ for rewetted soils (IPCC, 2014).

For analyzing $CH_4$, emissions, we fitted an exponential WTD response function according to Tiemeyer et al. (2020) and Evans
et al. (2021):

$$CH_4(WTD) = CH_{4\,min} + c * e^{-d*WTD} \qquad \text{(Eq. 3)}$$

where $CH_{4\,min}$ is the lower asymptote while c and d control the shape of the exponential function (Tiemeyer et al., 2020). The Danish sites, for which $CH_4$ emission data were available, represented drained and restored cropland and grassland, where the water level at least under experimental conditions was close to surface. Methane emission from ditches ($CH_{4\,ditch}$) were
estimated, by considering a fraction of the land, i.e., 10%, where drainage ditches are located. Given the applied grid size of 10 m, this corresponds to an averaged drainage ditch dimension of 1 m. The applied EFs for $CH_{4\,ditch}$ were 1,165 kg $CH_4$ ha$^{-1}$ yr$^{-1}$ for cropland and 948 kg $CH_4$ ha$^{-1}$ yr$^{-1}$ for grassland (IPCC, 2014).

Data for $N_2O$ emissions showed no systematic WTD dependence and in consequence, land use specific EFs were applied. We applied the EFs from Wilson et al. (2016) as updated from the IPCC (2014) wetlands supplement: 13.0 kg $N_2O$-N ha$^{-1}$ yr$^{-1}$ for
cropland, 4.7 kg $N_2O$-N ha$^{-1}$ yr$^{-1}$ as average for grassland (deep/shallow drained, nutrient rich/poor) and 0.1 kg $N_2O$-N ha$^{-1}$ yr$^{-1}$ for rewetted organic soils.

All GHGs were converted to $CO_2$-eq using their global warming potential over a 100 year period according to the 6th IPCC assessment report (Forster et al., 2021) where 1 kg $CH_4$ = 27 kg $CO_2$ and 1 kg $N_2O$ = 273 kg $CO_2$. For applying the land use specific EFs and WTD response functions, we used a 2020 land use classification for Denmark (Levin & Gyldenkærne, 2022).



Based on the available WTD observations, the WTD map captures a long-term average summertime condition. Since the applied GHG upscaling method is based on annual mean WTD, a scaling parameter is subtracted from the summertime WTD map to obtain an annual average. As described in section 2.2 the annual variability is estimated to be 0.25 m. In order to correct for seasonality, 0.125 m was subtracted from the summer WTD map. Negative values were set to zero.

## 3 Results

### 3.1 Water table depth model

The hyperparameter tuning of the GBDT model resulted in the following results: depth = 10, learning_rate = 0.05, subsample = 0.8, rsm = 0.8, min_data_in_leaf = 1 and l2_leaf_reg = 5. The GBDT model was trained against $WTD_t$, but throughout the manuscript results and analysis are based on the back transformed variable WTD. Figure 3 depicts the final simulated WTD map that represents a long-time average summertime condition for the period of 2000 to 2021 for Åmosen, which is one of the 200 largest peatlands in Denmark. For the visual assessment, a colour scheme that emphasizes the depth interval of 0 to 1 m has been selected. Even though WTD is simulated for a larger domain (lowland soils and river valleys), only grid cells with OC > 12% are shown, since the applied GHG upscaling method is only valid for such conditions. The WTD map discloses a distinct spatial heterogeneity with fully saturated conditions laying in very close vicinity to well drained conditions.

We applied SHAP to investigate feature importance of the trained GBDT model for the well and auger WTD observations, 205 i.e., excluding the dummy points (Figure 4). The six most important covariates were, ordered in high to low importance: horizontal distance to water bodies, horizontal distance to ditches, vertical distance to water bodies, wetness rank based on cropping history, clay content of the deepest soil horizon and land surface temperature (LST). Negative SHAP values are associated to negative impact, i.e., more shallow WTD and positive SHAP values are linked to a deeper water table. Locations close to water bodies exclusively possess negative SHAP values whereas locations with a large horizontal distance to the 210 closest water body have both negative and positive SHAP values. The SHAP values for the horizontal distance to drain ditches are separated with positive values (producing a deeper WTD) for low distances, which clearly reveals the functioning of the added dummy points representing well drained conditions along the drain ditches. The interpretation of the SHAP values for the vertical distance to water bodies is that WTD does not follow small scale topographical variation and instead the water table has a smoother variation than topography, which results in a deeper WTD (positive SHAP value) for areas with a high 215 vertical distance to the nearest water body. The wetness classes based on the cropping classes show a clear WTD sensitivity, where the low ranks, which are linked to crops that favor well-drained conditions possess positive SHAP values and the wet classes relate to a negative impact on the simulated WTD. A high clay percentage produces a positive impact on the prediction, i.e., deeper WTD. LST also shows a clear link to WTD, with higher values yielding a deeper WTD and lower LST resulting in more water saturated conditions.

Figure 3 exemplifies three of the seven listed covariates (i.e., LST, wetness rank and vertical distance to nearest water body) to elucidate key connections between model input and output. For LST (Figure 3b), there is a direct relationship, with lower



LST in areas of high saturation caused by evaporative cooling of the land surface due to high water availability, whereas deeper water tables, i.e., drier conditions, are collocated with higher LST. Further, we observe a good agreement between WTD and the wetness rank, derived from the cropping history from 2016 to 2020 (Figure 3c). Fields with crops associated with a wet

rank, i.e., permanent poor grassland with low nitrogen application rates, are associated with a low WTD, while crops that require drainage, e.g., winter wheat, potatoes or sugar beet, are found at fields with a deeper WTD. The agronomic requirements reflected by the ranked wetness map, are characterized by plausible mean WTD, which show consistent differences between each other. For the entire domain, the mean WTD values for the three wettest categories are below 0.4 m, whereas the three dry categories have a WTD of 0.65 m and deeper. Topographical variability is generally low in peatlands.

Nevertheless, the vertical distance to the closest waterbody reveals small-scale topographical features that effect the simulated WTD (Figure 3d).

Figure 5 depicts the spatial distribution of SHAP values for the same three covariates substantiates previous findings (Figure 4). High LST has a positive impact on the prediction resulting in deep WTD and low LST has a negative impact on the prediction. In the case of drained forest, which has a low LST, the negative impact of LST is overruled by the positive impact

of the crop based wetness rank. The latter shows a very clear separation of negative impact for the wet ranks and positive impact for the dry ranks. The negative impact of the vertical distance to the nearest waterbody is predominately limited to locations that are actually river or lake grids. A distinct positive impact is found for locations with a high vertical distance.

In order to assess the overall accuracy of the WTD map, we conducted a five-fold cross validation experiment. For this, five GBDT models were trained using 80% of the data for training and 20% of the data was held back for validation. The five

validation datasets were sampled so each WTD observation served exactly once as validation data. Figure 6 presents the scatter density plot of observed and simulated WTD both, a) including and b) excluding the dummy points, for the five validation datasets. The effect of the WTD transformation and the weighting scheme of WTD data in the depth interval of 0 to 0.5 m becomes apparent. The model shows the best accuracy for the shallow water table interval, whereas the performance deteriorates below a WTD of 1 m. The poor performance of WTD below 1 m can be explained by the transfer function which

hinders the GBDT model to discriminate WTD variability below a WTD of 1 m. In the case of WTD driven GHG upscaling, this is acceptable since the WTD response functions are not sensitive to changes in WTD deeper than approx. 0.5 m. The scatter plot reveals that the dummy points with zero and negative WTD, i.e., above terrain, are generally represented quite well by the GBDT model. Taking only the well and auger WTD observations into consideration, a slight bias for the shallow WTD observations becomes evident.

Table 3 quantifies the performance of the GBDT model for the five-fold cross validation test both including and excluding the dummy points. The bias of the top 0.5 m interval and the 0.5 to 1 m interval was -0.05 m and 0.08 m, respectively. The performance for the well and auger subset is poorer for the top 0.5 interval with a bias of -0.2 m. For the deeper intervals the metric scores are comparable for the entire trainingsdataset and the subset based on exclusively well and auger data. WTD deeper than 1 m perform worst on all stated metrics, which underpins the visual assessment of Figure 6.



For the entire model domain, the GBDT model predicts a WTD interval sensitive to GHG variability, i.e., below 0.5 m, for 36% of the area. For the delineated peat soils, with OC >12%, this area amounts to 54%. After correcting from summer to annual conditions, i.e., subtracting 0.125 m (half the mean annual amplitude), these area estimates increase to 45% and 64%, respectively. For agricultural areas with OC >12% the mean WTD is 0.49 m with a standard deviation of 0.35 m which underpins the distinct WTD variability in peatlands, also within the range of WTD associated with high sensitivity of the resulting emissions of $CO_2$, $CH_4$ and $N_2O$.

### 3.2 Danish greenhouse gas response functions

The parametrisation of the fitted WTD driven response functions for $CO_2$ and $CH_4$ emissions (Figure 7) showed a systematic relationship where $CO_2$ emissions increased with increasing WTD between 0 and 0.4–0.5 m before reaching an asymptotic level of 10 Mg $CO_2$-C ha$^{-1}$ yr$^{-1}$. The fitted parameters are as follows: $CO_2$-$C_{min}$ = 1.132, $CO_2$-$C_{diff}$ = 10.903, a = 6.415 and b = 14.183. $CH_4$ emissions were consistently negligible at WTD depths below 0.2–0.3 m, but increased at higher WTD to emissions of up to 0.8 Mg C ha$^{-1}$ yr$^{-1}$. However, it is clear that a shallow WTD does not necessarily cause high $CH_4$ emissions, but rather provides a window of opportunity for methane fluxes to the atmosphere. The fitted parameters are as follows: $CH_{4\ min}$ = -21.48, c = 258.83 and d = -5.16. $N_2O$ emissions were not modelled, but average values for observations at WTD >0.3 m (n = 19) and <0.3 m (n = 6) were 13.3 and 3.8 kg N ha$^{-1}$ yr$^{-1}$, respectively, thus representing a magnitude similar to IPCC EFs, although somewhat higher as compared to the rewetted category (Wilson et al., 2016).

### 3.3 Upscaled greenhouse gas emissions

GHG emissions can be estimated based on the following, (1) the GHG upscaling method presented in section 2.5, (2) the long-term annual average WTD map, (3) a land use map and (4) a map delineating the drainage ditches. With a spatial resolution of 10 m, the WTD maps open the possibility to estimate GHG at equally high resolution. However, given the apparent uncertainties in the WTD map as well as in the WTD response functions, GHG are aggregated to national scale.

For the approximately 74,000 ha with OC >12%, the total emission of the three gasses $CO_2$, $N_2O$ and $CH_4$ amounts to 2.6 Mt $CO_2$-eq. Figure 8 depicts the relationship between WTD and the estimated GHG emissions, expressed as emission factor converted to $CO_2$-eq. Emissions of $CO_2$ dominate the GHG budget at WTD deeper than approx. 0.1 m, whereas methane emissions become dominating at WTD closer to the soil surface. The contribution from $CH_4$ emissions is apparent, starting from a WTD of approx. 0.3 m, whereas low $CH_4$ emissions for deeper WTD are related to the minor $CH_{4\ ditch}$ component which is not WTD dependent and takes place throughout the lowland soils. $N_2O$ emissions have no WTD response function and thus, emissions are rather constant across the WTD variability. Spatial heterogeneity of $N_2O$ emissions is based on the applied land use specific emissions factors which may indirectly be linked to WTD variability, which results in a slight decrease of $N_2O$ emissions with decreasing WTD. Based on the minimum of total GHG emissions shown in Figure 8, a WTD of approx. 0.04 m can be identified as the optimal WTD for minimal GHG emissions, i.e., 5.6 $CO_2$-eq Mg ha$^{-1}$ yr$^{-1}$. Yet, the exact numbers should be viewed as indicative, since for example the possibility of negative $CO_2$ emissions at low WTD depends on the





presence of wetland vegetation. Nevertheless, even in the absence of negative $CO_2$ emissions, the data indicates that an optimal rewetting strategy should aim at a WTD at a range between 0 and 0.1 m WTD to balance the trade-off between $CO_2$ and $CH_4$ emissions.

Table 4 states the emission factors for the three considered land use classes. The emission factors for $N_2O$ are in direct agreement with the ones stated in section 2.5 whereas the emission factors for $CO_2$ and $CH_4$ are affected by the modelled WTD variability. In total, based on area, but also emission factor, cropland dominates the GHG emissions of peatlands in Denmark. As expected, emission factors from rewetted peat soils are lowest.

### 3.4 Rewetting scenarios

The combination of a high-resolution WTD map and WTD response functions of GHG emissions allows to evaluate the effects of alternative rewetting scenarios. For this, it is assumed that the WTD of an agricultural field can be changed to the optimal WTD, allowing a reduction of total GHG emissions to 5.6 $CO_2$-eq Mg ha$^{-1}$ yr$^{-1}$ (Figure 8). Figure 9 shows the results of three rewetting scenarios and a theoretical baseline as functions of the peatland area that is rewetted with a maximum of 74,000 ha. The baseline expresses the theoretical maximum emission reduction with the assumption that all agricultural fields are

originally well drained and thus having a uniform reduction of 36.8 $CO_2$-eq Mg ha$^{-1}$ yr$^{-1}$, which expresses the reduction from the maximum asymptote to the minimum of the total GHG curve in Figure 8. In the three rewetting scenarios the reference emissions are derived from the WTD response functions and thus, many agricultural fields have a lower reference emission than the baseline, which will result in a decreased emission reduction with respect to the baseline. The first scenario prioritises wet fields in the restoration, i.e., the agricultural fields with the lowest mean WTD are prioritised in the rewetting strategy.

This scenario can be regarded pessimistic with respect to the expected GHG emission reduction. The second scenario prioritises the dry fields with the lowest mean WTD, which in turn can be considered an optimistic scenario. The third scenario selects fields in random order and lies in between the optimistic and the pessimistic scenarios. The prioritization order is based on over 79,000 digitized fields (Figure 3) and the WTD of an entire field is set to 0.04 m for calculating the reduction in GHG emissions. Also, we have assessed the sensitivity of the applied correction from summer WTD to annual WTD by running the

scenarios with the corrected (annual) and the uncorrected (summer) WTD maps. Since WTD is lower for summer conditions, the estimated reduction of GHG emissions is higher than the estimate based on annual WTD. In case the entire peatland area is rewetted the reduction in GHG emissions is estimated to be 2.0 $CO_2$-eq Mt yr$^{-1}$ by all three scenarios (2.3 $CO_2$-eq Mt yr$^{-1}$ using the summer WTD map). However, we observe large discrepancies between the restoration scenarios if only a fraction of the total peat area is rewetted. Prioritising dryer fields provides a high reduction already starting with the first rewetted fields

whereas prioritising wet fields shows little reduction. In fact, the reduction potential in the wet scenarios is just 30% of the dry scenario for the first 10,000 ha. A deviation of 50% between wet and dry scenario is first exceeded for a rewetting area of above 20,000 ha. The random scenario lies in between the optimistic and pessimistic scenarios with a linear emission reduction of 26.1 Mg $CO_2$-eq ha$^{-1}$ yr$^{-1}$. The reduction factor of the random scenario is 10.7 Mg $CO_2$-eq ha$^{-1}$ yr$^{-1}$ lower than the baseline



scenario. This relates to the WTD map that introduces spatial variability in the random scenario opposed to the fully drained
conditions assumed in the baseline scenario.

## 4 Discussion and conclusion

Machine learning can utilize the broad spectrum of geo-environmental big data to model WTD at national scale of Denmark
with a reasonable accuracy, taking the quality of available WTD observations into consideration. The five-fold cross validation
experiment revealed an acceptable residual variance for the most shallow WTD interval (MAE of 0.08 m). Despite all efforts
to finetune the GBDT model to perform well for the shallow WTD interval, a considerable residual variance (MAE of 0.27 m)
was evident when only taking the well and auger WTD observations into consideration. As a consequence of the applied WTD
transformation, performance decreased substantially for the deeper WTD. Similar findings were documented by Bechtold et
al. (2014), which are mainly related to the applied WTD transformation. However, several sources of uncertainties remain to
be addressed, such as the difficulties to model a long-term average WTD based on a heterogenous training dataset containing
observation from summer months from different years. Future work should aim at homogenizing WTD observations, e.g. by
normalizing to climate variability, to derive a more representative training dataset. Nevertheless, we believe that the
comprehensive training dataset provides meaningful information to the GBDT model to predict an average summertime
condition.

The SHAP analysis revealed that topography, water body proximity and land use were the most important covariates in the
trained GBDT model. Similar findings were reported by other WTD ML-based modelling studies (Bechtold et al., 2014; Koch
et al., 2019). The sign and magnitude of the SHAP values provided detailed knowledge on how covariates are linked to WTD.
Similar findings have been obtained by Bechtold et al. (2014) applying partial dependence plots. In contrast to Koch et al.
(2019, 2020), who modelled WTD over the entire land phase of Denmark, we found geology related covariates less informative
for modelling exclusively peat soils. It remains unresolved if this relates to poorer quality of lithological and geological
information in peatlands or if peatland hydrology processes are predominately controlled by topography and waterbody
proximity.

The Gompertz parametrization of the WTD response function for $CO_2$ for Danish organic soils (Figure 6) was strikingly similar
to the parametrization based on a larger German dataset (Tiemeyer et al., 2020). Hence, applying the parameters from Tiemeyer
et al. (2020) in our upscaling study resulted in $CO_2$ emissions that deviated by just 1% on average with respect to the $CO_2$
emissions based on the Danish Gompertz parametrization. This underlines the strong and consistent effect of WTD as a driver
of $CO_2$ emissions from organic soils across climatic and agroecological conditions. Similar conclusions were reached when
comparing our parametrization of the $CH_4$ response function with the German parametrization (Tiemeyer et al., 2020).

Although, supported by the present study and Tiemeyer et al. (2020), the asymptotic WTD response curve for $CO_2$ emissions
may not be universally applicable. Evans et al. (2021) analysed $CO_2$ emissions, based on published eddy covariance studies
on boreal and temperate peatlands, and suggested a linearly increasing emission with increasing WTD. The Danish $CO_2$ data





presented here are predominately in the linear range of the Gompertz function (0 – 0.5 m). Future research should target measuring GHG emissions at sites with a thick unsaturated peat soil to investigate how the response function can be extrapolated to deeper WTD. Nevertheless, if taking the actual peat depth into consideration, which can be considered a lower boundary of the response functions, applying the Evans et al. (2021) model may provide comparable results in the present

analysis. However, a high-quality peat depth map is required to substantiate this statement. Both studies provide similar findings for the shape of the $CH_4$ response function, which can be further substantiated by the Danish flux data.

$N_2O$ emissions factors were not updated by our study, and instead IPCC emission factors were used (Wilson et al., 2016). However, the synthesized Danish $N_2O$ data presented herein suggested average emissions for observations at WTD >0.3 m (n = 19) and <0.3 m (n = 6) to be 13.3 and 3.8 kg N ha$^{-1}$ yr$^{-1}$, respectively. These figures are very comparable to the ones presented

by Wilson et al. (2016).

The official Danish national inventory has reported an emission of 3.00 Mt $CO_2$-eq from soils with OC >12% (Nielsen et al. 2022), which is 15 % higher than our estimate. The green deal in Denmark was guided by the Danish council on climate change who conducted estimations of the potential reduction in GHG emissions as a consequence of large-scale peatland restoration (Klimarådet, 2020). For peatland soils with OC > 12%, a reduction potential of 2.71 mil. Mg $CO_2$-eq yr$^{-1}$ was assessed. Our

findings (2.0 Mg $CO_2$-eq ha$^{-1}$ yr$^{-1}$) are considerably lower, and the difference can be attributed to the fact that our results are based on a lower baseline which considers the WTD map instead of assuming fully drained conditions. The distinct difference of 31% between the fully drained baseline and the WTD driven reduction potential is also clearly visible in Figure 9. Reflecting on the assessed restoration scenarios, it may be assumed that the "prioritized wet" scenario is most realistic, since this scenario prioritises marginal wet fields of low economic value in the restoration order. This sheds a pessimistic outlook on the mitigation

potential when only restoring a fraction of entire peatland. At the same time it emphasizes the value our framework which can guide peatland restoration to be most effective.

Many of the agricultural organic soils in Denmark have been drained for years. As stated in Greve et al. (2014), a substantial loss in the area qualifying as OC > 12% is recorded. The organic soil map by Greve et al. (2014) was created based on measured data in 2010 with a definition of a minimum depth of 0.3 m organic layer, which resulted in the delineation of the 74,000 ha

with OC > 12% used in this study. A further reduction in the area with organic soils with this minimum definition is likely to have occurred. As a consequence, it is disputable that the Gompertz function can be applied to all the currently reported 74,000 hectares. Thus, the WTD function should only be used for those cases where the WTD is in the organic layer. If the WTD is deeper (e.g., in a sand layer) then the depth of the peat should be the lower boundary to derived an effective 'WTD' to be used in the model. Therefore, combining the present data analysis with a map of peat depth at national scale would provide a further

step towards a consolidated estimate of GHG emissions from Danish organic soils.

The applied WTD response functions for $CO_2$ and $CH_4$ yield emissions at a scale corresponding to the applied WTD map. In our case, the 10 m resolution of the WTD map provides high-resolution GHG estimates that could allegedly support sub-field restoration projects. Taking all uncertainties into consideration, we do not support such a spatially differentiated application. However, due to the non-linearity of the response functions and the distinct spatial heterogeneity of WTD, an initial high-



resolution assessment is required before aggregating the results. Future work should address the relationship between scale and uncertainty of the proposed GHG upscaling framework to identify the representative scale at which the upscaling model has a potential for obtaining a predictive accuracy corresponding to a given acceptable accuracy (Refsgaard et al., 2016).

Our restoration scenarios (Figure 9) only comprise a rewetting of the domain and, in fact, peatland restoration is a much wider management term that covers various ecosystem services, such as biodiversity and nutrient retention (Andersen et al., 2017;

Hambäck et al., 2023). Thus, peatland restoration is not exclusively targeting climate change mitigation with a broad suite of measures.

We draw the following main conclusion from out work.

- The WTD model reveals that 64% of the Danish peatland with OC > 12% has a WTD in the depth interval sensitive to GHG emission (0 – 0.5 m).
- The fitted WTD response functions and emission factors for Danish conditions are in good agreement with the international literature.
- The 74,000 ha farmed peatland with OC > 12% emit 2.6 Mt CO2-eq, which is 15% lower than the officially reported national emission in 2020. A total rewetting would decrease the GHG emissions by 77%.
- The order in which peatland is rewetted has substantial implications for the expected GHG reduction and well drained
400       fields should be prioritized to achieve the highest effect.

**Code & Data availability**

The water table depth map is made freely available via the following repository: https://doi.org/10.22008/FK2/0AFGQT All code and supporting data will be made available upon request.

**Author contribution**

JK and SS designed the water table depth model and the rewetting scenarios. JK developed the code for the water table depth model and conducted the formal analysis. MG, SG, CH, GL, SW and LE contributed with data to the analysis. MG and CH contributed with water table depth data, SG and GL contributed with covariate data and SW and LE compiled the Danish GHG data synthesis and fitted the water table depth response functions. JK prepared the manuscript with contributions from all co-authors.

**Acknowledgements**

The authors would like to thank the financial support by the Danish Ministry of Climate, Energy and Utilities.



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

Derivation of greenhouse gas emission factors for peatlands managed for extraction in the Republic of Ireland and the United



Kingdom, Biogeosciences, doi:10.5194/bg-12-5291-2015, 2015.

Wilson, D., Blain, D., Couwenberg, J., Evans, C. D., Murdiyarso, D., Page, S. E., Renou-Wilson, F., Rieley, J. O., Strack, M.

and Tuittila, E. S.: Greenhouse gas emission factors associated with rewetting of organic soils, [online] Available from:
http://hdl.handle.net/10012/11532, 2016.





## List of figures



**Figure 1:** a) map of Denmark, b) including the island Bornholm, indicating the entire domain of the WTD model (lowland & river valley) and the focus area with organic content (OC) > 12%. c) zoom into Åmosen with different sources of WTD data. The WTD data sources are not differentiated in a) and simply shown as black dots. Location of c) shown in a). d) overview figure indicating the location of Denmark in Northern Europe.




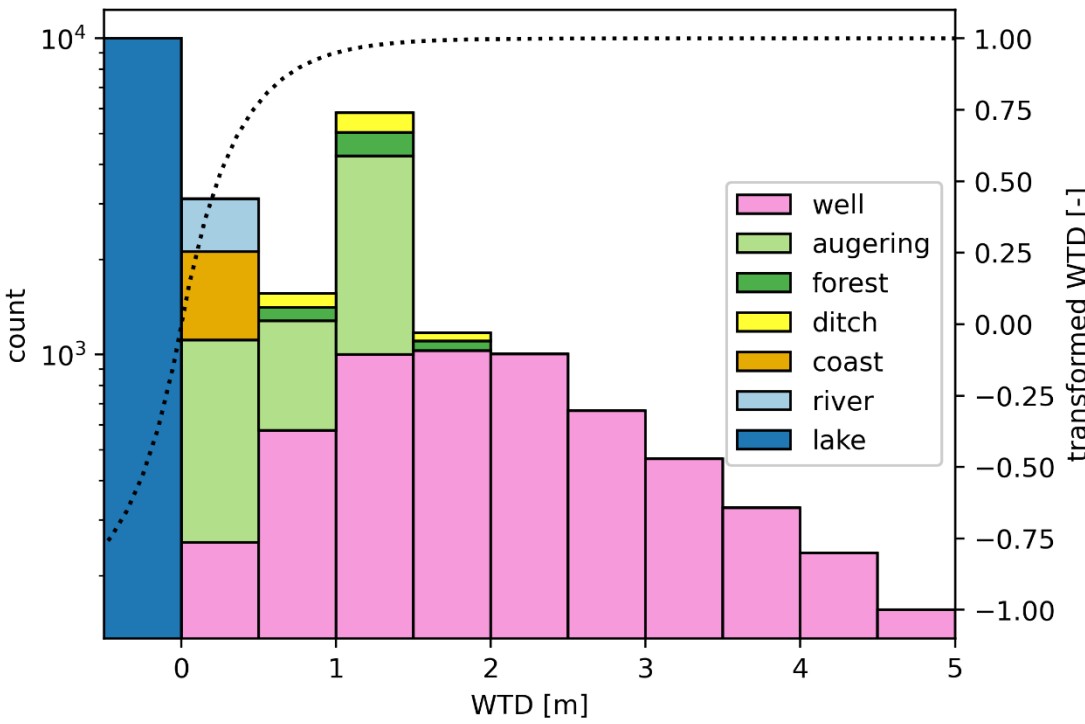

**Figure 2: Variability and frequency of the analysed WTD data with respect to their sources. Dashed line indicates the transfer function yielding transformed WTD (WTD$_t$) on the secondary y-axis.**

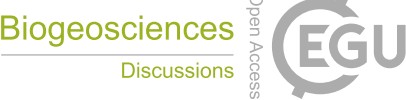

**Figure 3: a) The modelled WTD map at 10 m resolution for Åmosen, same zoom as Figure 1 c). Three key covariates: b) land surface temperature (LST), c) wetness rank derived from the 2016 to 2020 cropping history and d) vertical distance to nearest water body (river, lake or coast). Polygon showing the delineation of agricultural fields is added to all panels.**



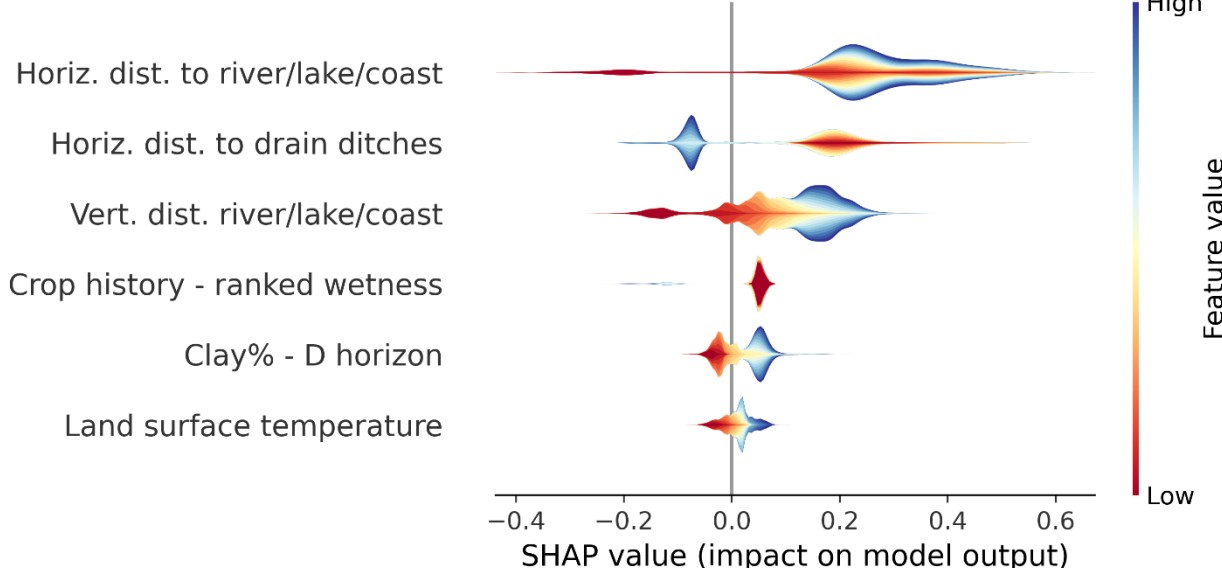

**Figure 4: SHAP values for the six most important covariates based on an analysis using all well and auger WTD observations. The SHAP value interprets the covariate's impact on the prediction. The violine plots are colour coded based on the stacked values of the covariates and the height indicates the density of the data.**


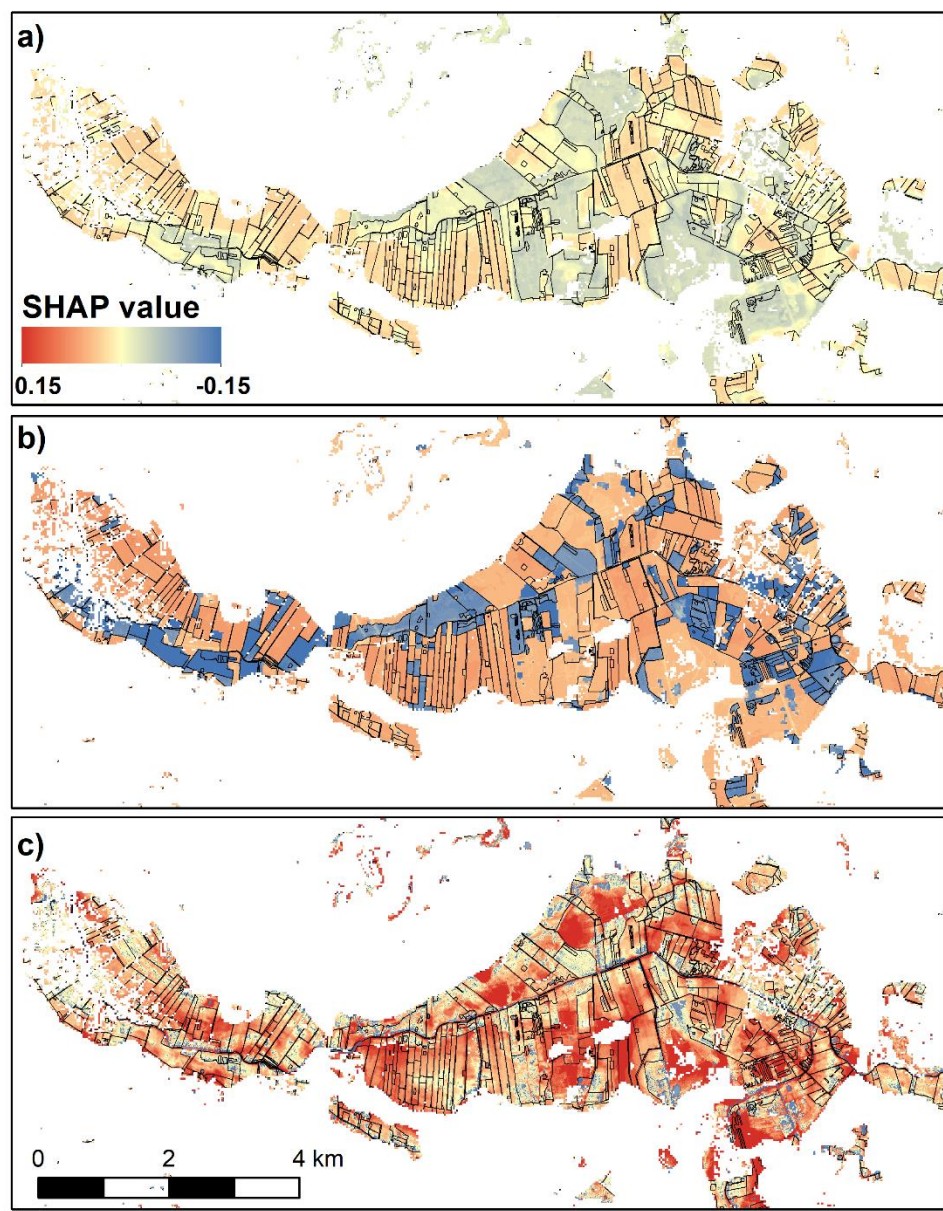

**Figure 5: SHAP values for the prediction dataset shown for three selected covariates: a) LST, b) cropping history- ranked wetness and c) vertical distance to nearest waterbody. Results are shown for Åmosen, same zoom as Figure 1 c) and Figure 3.**





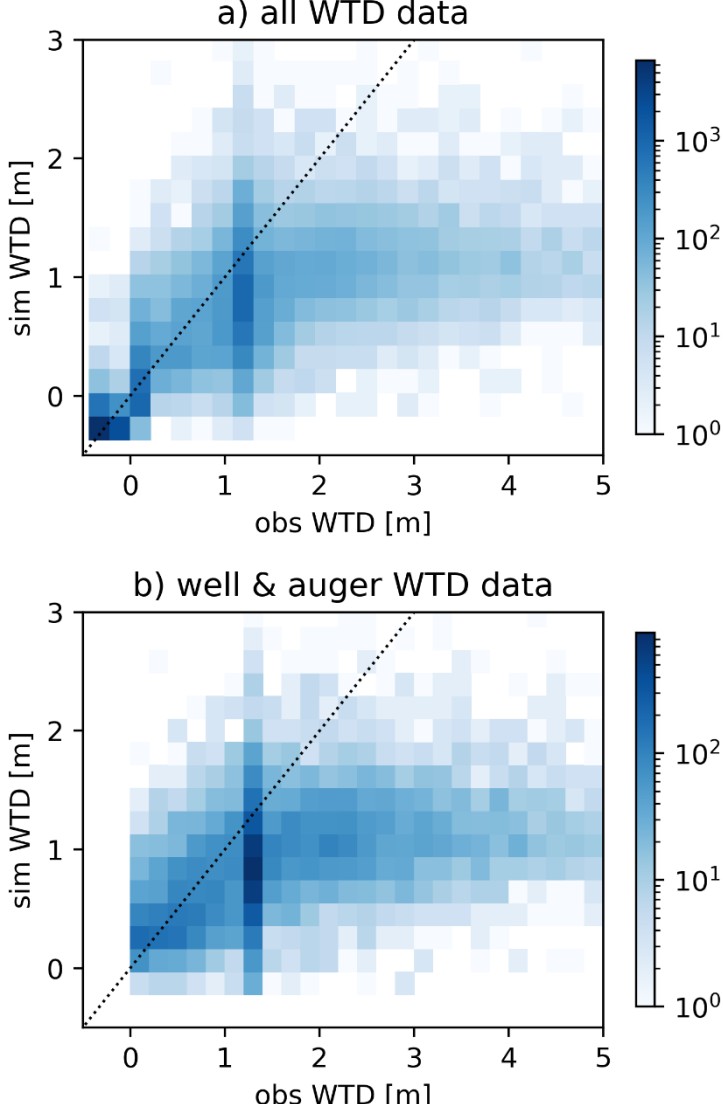

Figure 6: Density scatter plots, with 0.2 m bins, for the applied five-fold cross validation test. Dashed line represents the 1:1 line between observed (obs) and simulated (sim) WTD. The colourbar indicates the data count for each bin. In a) all WTD data are plotted and in b) only a subset containing the well and auger observation are shown.





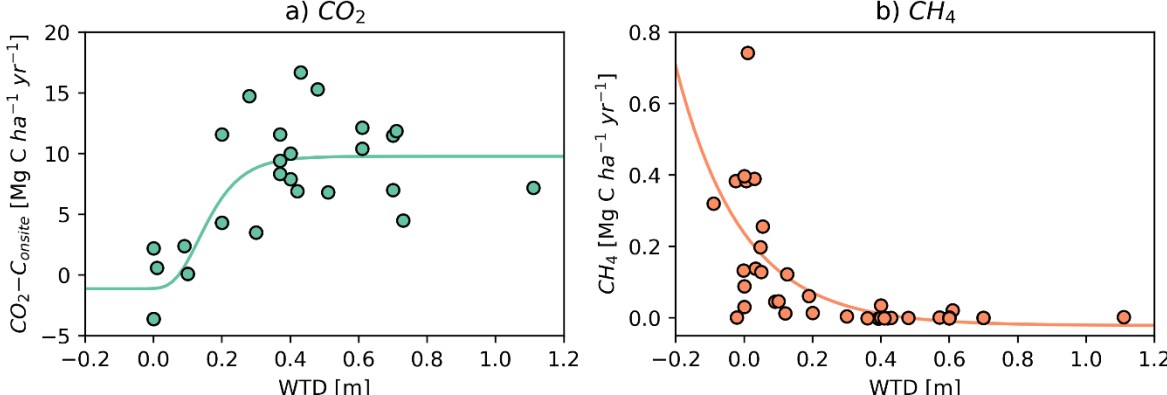

**Figure 7: Annual net ecosystem carbon balance of CO₂ (left panel) and emissions of CH₄ (right panel) in Danish organic soils plotted against mean water table depth (WTD). Sources of data are shown in Supplementary Table S1.**






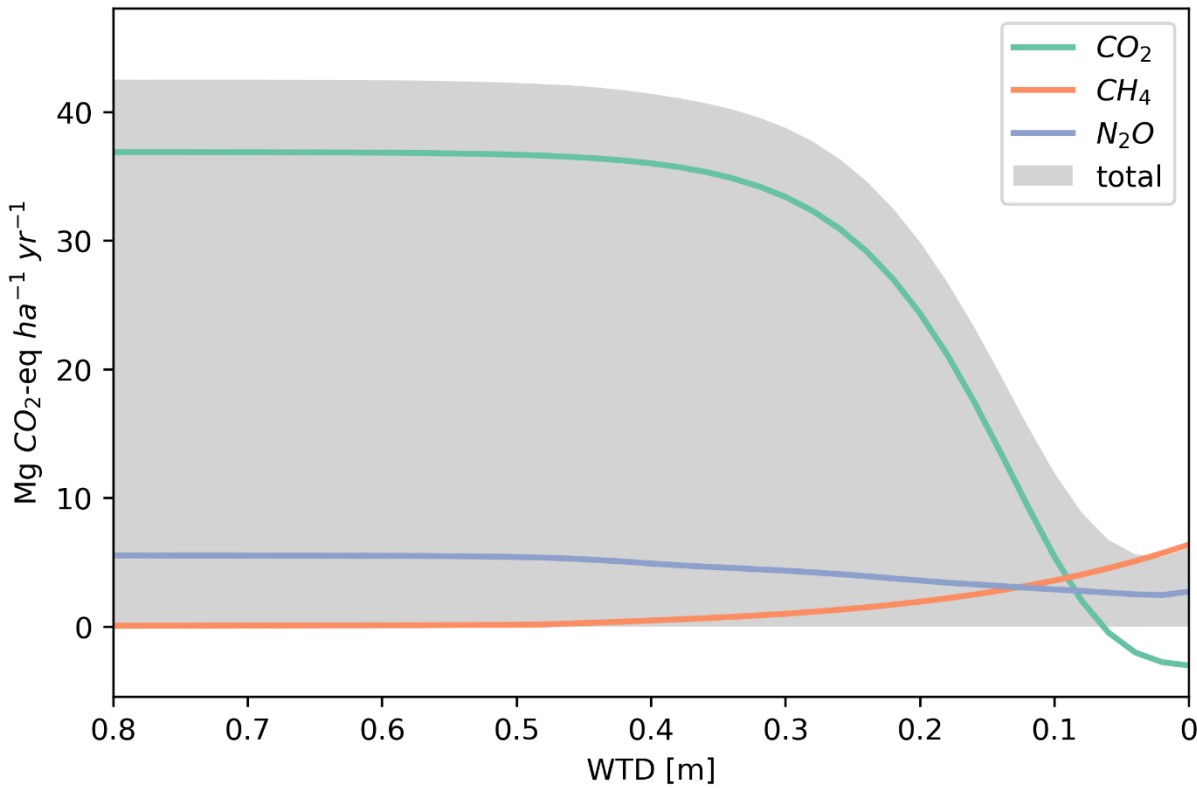

**Figure 8: The relationship between greenhouse gas emissions (CO₂, CH₄ and N₂O) and WTD. The emissions are stated in CO₂ equivalents applying 100 year global warming potentials: 1 kg CH₄ = 27 kg CO₂ and 1 kg N₂O = 273 kg CO₂.**





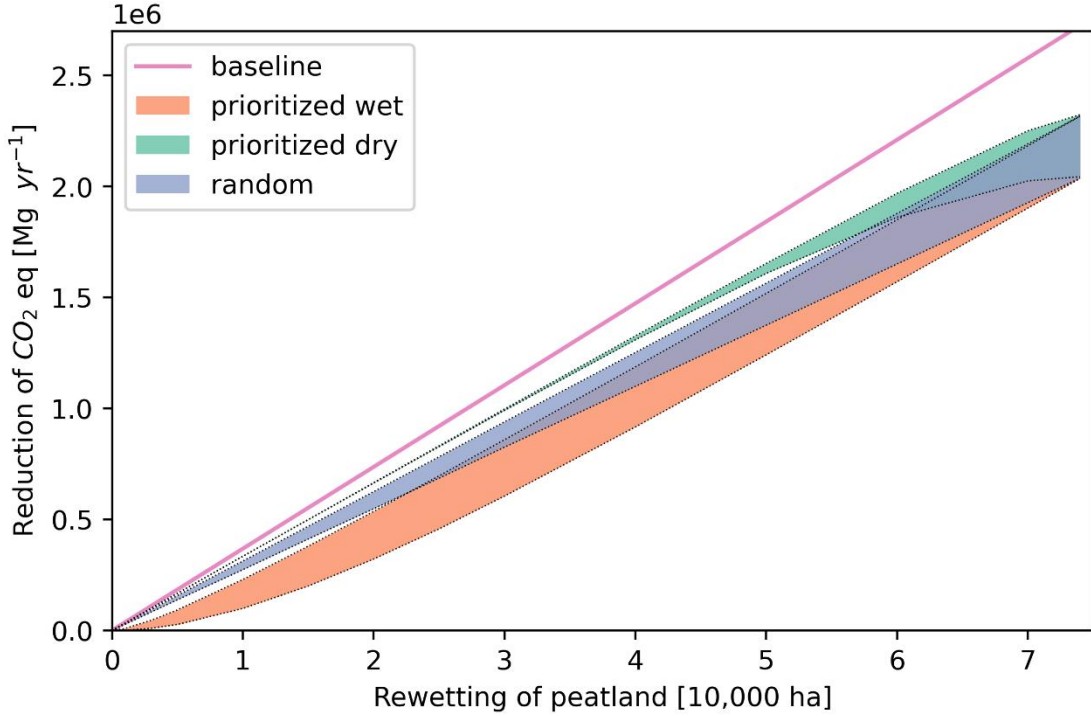


**Figure 9: The estimated reduction of GHG emissions in CO₂ equivalents as a function of rewetted peatland. Four scenarios are tested to investigate the potential mitigation effect of alternative rewetting strategies. The uncertainty bands relate to the utilization of summer WTD (upper potential) and annual WTD (lower potential).**



**List of tables**

**Table 1: WTD observations used as training data with information on number of data (n), WTD mean and standard deviation (std). Source marked with asterisk represent dummy points.**

| Source | n | mean [m] | std [m] |
|--------|------|----------|---------|
| Coast* | 1000 | 0 | 0 |
| Ditch* | 1000 | 1.21 | 0.2 |
| Forest* | 1000 | 1.21 | 0.2 |
| Lake | 9980 | -0.25 | 0.05 |
| River* | 1000 | 0 | 0 |
| Cores | 4796 | 0.96 | 0.39 |
| Well | 5716 | 2.12 | 1.09 |

**Table 2: Covariates used in the WTD model. Covariates marked with asterisk are categorical.**

| Category | Covariate | Source |
|----------|-----------|--------|
| Topography & water body proximity | Elevation | Digital Elevation Model (2018) provided by the Danish Agency for Data Supply and Infrastructure (SDFI) |
| | Vertical distance to river/lake/coast | |
| | Horizontal distance to river/lake/coast | |
| | Vertical distance to drain ditches | |
| | Horizontal distance to drain ditches | |
| | Terrain slope | |
| Lithology | Clay percentage in 4 soil horizons | Adhikari et al. (2013) |
| | Clay thickness | Stisen et al. (2019) |
| | Soil map* | Pedersen et al. (2011) |
| | Landscape types* | Madsen et al. (1992) |
| | Organic content | Greve et al. (2019) |
| Landsat | Land surface temperature | Potapov et al. (2020) |
| | Normalized difference vegetation index | |
| | Normalized difference water index | |
| | Modified normalized difference water index 1 | |
| | Modified normalized difference water index 2 | |
| Sentinel 1 | Global backscatter model - vv | Bauer-Marschallinger et al. (2021) |
| | Global backscatter model - vh | |
| | Water & wetness | Copernicus |
| Land Use | Land use map* | Parente et al. (2021) |
| | Cropping history - ranked wetness | Aarhus University - DCE |
| | Degree of urbanization | SDFE |
| | Forest - wet/dry | Levin (2019) |
| Hydrological model | mean summer WTD at 100 m | Henriksen et al. (2020) |





**Table 3: Performance of the five-fold cross validation test for three depth intervals assessed by three metrics: Mean error (ME), mean absolute error (MAE) and root mean squared error (RMSE). Only well and auger WTD observations were considered for metric scores stated in brackets.**

| Interval [cm] | ME [m] | MAE [m] | RMSE [m] |
|---|---|---|---|
| 0 - 50 | -0.05 (-0.20) | 0.08 (0.27) | 0.16 (0.38) |
| 50- 100 | 0.08 (0.07) | 0.31 (0.31) | 0.39 (0.39) |
| > 100 | 0.79 (0.85) | 0.85 (0.91) | 1.15 (1.23) |

**Table 4: The implied emission factors for $CO_2$-$C_{organic}$, $CH_4$ $_{organic}$, $N_2O$-$N_{organic}$ and total greenhouse gas (GHG) emissions applying 100 year global warming potentials: 1 kg $CH_4$ = 27 kg $CO_2$ and 1 kg $N_2O$ = 273 kg $CO_2$. The applied land use map is derived from Levin and Gyldenkærne (2022).**

| Land use | Area [ha] | $CO_2$-$C_{organic}$ [Mg C ha$^{-1}$ yr$^{-1}$] | $CH_4$ $_{organic}$ [kg $CH_4$ ha$^{-1}$ yr$^{-1}$] | $N_2O$-$N_{organic}$ [kg N ha$^{-1}$ yr$^{-1}$] | GHG [$CO_2$-eq ha$^{-1}$ yr$^{-1}$] |
|---|---|---|---|---|---|
| Cropland | 56249 | 9.4 | 19.5 | 13.0 | 40.3 |
| Grassland | 11238 | 7.4 | 59.2 | 4.7 | 30.6 |
| Rewetted | 4904 | -0.3 | 193.7 | 0.1 | 4.1 |