# Peer review of "Water table driven greenhouse gas emission estimate guides peatland restoration at national scale"

_Biogeosciences, 2023_

## Referee Comment (RC1)

**Referee comments on "Water table driven greenhouse gas emission estimate guides peatland restoration at national scale" – bg-2023-23**

Within the article the authors aim to calculate national peatland greenhouse gas (GHG) emissions for Denmark and potential rewetting emission reductions following various rewetting strategies. The approach consists of three steps: (1) Modelling average water table depth (WTD) statistics for organic soils, (2) fitting a relation between WTD and GHG observations and (3) estimating total GHG emissions and (4) exploring emissions reduction pathways for Denmark.

The authors bridge multiple research disciplines, outcomes and approaches and present a clear and valuable narrative. The results could directly support peatland management decision-making processes aiming to minimize the contribution of the Danish peatlands to climate change. As the article might impact peatland management on a national scale it is important to address shortcomings and discuss results with nuance, which is something that the authors clearly do. However, I have several additional concerns, which I listed below. I recommend that these concerns are addressed before publication of the manuscript.

*Major concerns:*

*Long term average summer WTD*

The authors used WTD data to train the machine learning model for predicting average summer WTD for the complete Danish lowland/peatland areas. More or less half of this dataset originates from median well measurements. Besides, almost half of this WTD dataset originates from soil auger campaigns that consisted of one or two auger measurements. It is not clear from the text what methodology was used for determining the auger-campaign-WTD; (1) were oxidation marks used that reflect historic WTD characteristics or (2) was the WTD measured directly? In the latter case, summer WTD may fluctuate substantially and an auger WTD measurement is highly dependent on the timing measurement, and therefore the question remains if it is really valid to include this data within the research. To demonstrate the accuracy or uncertainty that is introduced, I suggest that the authors compare summer WTD averages of wells positioned nearby auger measurements.

Furthermore, it is unclear how many wells were actually positioned within in peat soils (OC > 12%). Are the well measurements representative enough for peatland WTD's, or mainly located within the lowland valleys?

The model was assigned to predict a draining effect of ditches by setting draining dummy points, which were found to function successfully within the SHAP analysis (Line 210). However, the authors did not discuss the spatial occurrence of ditches and explain why a ditch WTD of 1.21 (Line 101) would be a valid assumption. Are ditches always dry and thus draining the soils? Or do they also supply water during dry conditions? Is this more or less the same throughout the country?

*Model performance and ecosystem carbon balance*

A mean error of the trained model of -0.20 m for WTD estimates between 0-0.5 m (Table 3) seems quite high, given the high sensitivity of $CO_2$ emissions predicted for this WTD range (Fig. 7a) and the high percentual area with WTD within this range (Line 257). This seems problematic within further analysis. The authors are open about these high uncertainties (i.e. Line 275) and proceed the analysis

on a national scale. The mean error of -0.20 m means that the modelled WTD is more shallow as compared to the observed WTD. Would this mean that national estimates of total emissions are highly underestimated, and that emission reductions are underestimated as well? The MAE of 0.08 m (Line 324, Table 3) is based on validation including dummy points. As these points represent a large portion (1/3) of the dataset the model performance statistic seems biased. Therefore, I would suggest to exclude dummy points by default in Table 3 (and refer to these in text) and report performance statistics including dummy points between brackets instead.

It is exciting that the authors found very similar parameterization for the adopted Gompertz function used in Tiemeyer et al. (2020) (Fig. 7a) (the function to estimate NECB based on WTD). However, the amount of net ecosystem carbon balance (NECB) datapoints in the sensitive WTD range is quite low. As the authors already address, many linear relations have been fitted between WTD and NECB within literature. Except for one observation with a WTD > 1 m, a linear relation would not misfit the data in this research. Such a less steep linear relation would distribute the weight away from the currently very sensitive (and steep) 0- 0.5 m WTD-region and result in carbon balance estimations (and emission reduction estimations) that are less vulnerable to apparently quite substantial errors in WTD estimations (especially for 0-0.5 m WTD). I suggest that the authors conduct two sensitivity analyses: (1) the effect of water table bias on GHG emissions, and (2) the effect of Gompertz function on the GHG emissions by stretching, squeezing and using a linear relation.

*Covariate analysis*

In the results the authors discuss the impact of land surface temperature (LST) independently of ranked wetness. However, to what extent are these two covariates independent? A crop with deep WTD generally evaporates a higher amount of water, leading to more latent cooling. I would suggest to check the independency of the covariates. Furthermore, the small SHAP values for LST might not justify the high amount of attention in the results section (i.e. Fig. 3 and 5, from Line 228 onwards). In addition, the lower LST in areas of high saturation might not only be caused by more evaporative cooling, but maybe also due to induced heat transport towards deeper soil layers (wet peat soils conduct heat much better).

*Total GHG emissions and reduction pathways*

Given the urgency of climate action, I would suggest to include 25 year global warming potential to calculate total GHG budgets complementary to the 100 year timeframe that is currently applied in Fig. 8 and 9. In this case, the research would address both the short- and long-term effects of rewetting strategies.

*Other remarks:*

- I would suggest aligning the x-axis of Fig. 4 and the legend of Fig. 5 in order to facilitate a constant direction of thinking (i.e. negative SHAP values are related to shallow WTD).
- Figure 6 gives insight in the general performance of the GBDT models. I suggest to additionally show how well the models perform within the most important and sensitive WTD range of 0-1.0 m (maybe with grids of 0.1 m or 0.05 m).

- It is unfortunate that peat depth could not be taken into account within the analysis (Line 355, 376). Including such data would likely result in a lower prediction of emissions, which could be stated more directly in the text.

---

## Author Response (AR1)

**"Water table driven greenhouse gas emission estimate guides peatland restoration at national scale" – bg-2023-23**

[Reviewer comments in *italic blue font*; author replies normal font]

*Within the article the authors aim to calculate national peatland greenhouse gas (GHG) emissions for Denmark and potential rewetting emission reductions following various rewetting strategies. The approach consists of three steps: (1) Modelling average water table depth (WTD) statistics for organic soils, (2) fitting a relation between WTD and GHG observations and (3) estimating total GHG emissions and (4) exploring emissions reduction pathways for Denmark.*

*The authors bridge multiple research disciplines, outcomes and approaches and present a clear and valuable narrative. The results could directly support peatland management decision-making processes aiming to minimize the contribution of the Danish peatlands to climate change. As the article might impact peatland management on a national scale it is important to address shortcomings and discuss results with nuance, which is something that the authors clearly do. However, I have several additional concerns, which I listed below. I recommend that these concerns are addressed before publication of the manuscript.*

**Reply:** We would like to thank the reviewer for their very fair, thoughtful, and constructive review of our submission. We are glad that the novelty and relevance has been generally acknowledged by the reviewer. The reviewer states several very relevant points. Therefore, we will gladly pick them up during the revision to enrich the discussion of the manuscript. We totally agree, this article might impact the peatland management at national scale and uncertainties and shortcomings need to be addressed and discussed. Please find our point by point replies and concrete plans for the revision below.

*Major concerns:*

*Long term average summer WTD*

*The authors used WTD data to train the machine learning model for predicting average summer WTD for the complete Danish lowland/peatland areas. More or less half of this dataset originates from median well measurements. Besides, almost half of this WTD dataset originates from soil auger campaigns that consisted of one or two auger measurements. It is not clear from the text what methodology was used for determining the auger-campaign-WTD; (1) were oxidation marks used that reflect historic WTD characteristics or (2) was the WTD measured directly? In the latter case, summer WTD may fluctuate substantially and an auger WTD measurement is highly dependent on the timing measurement, and therefore the question remains if it is really valid to include this data within the research. To demonstrate the accuracy or uncertainty that is introduced, I suggest that the authors compare summer WTD averages of wells positioned nearby auger measurements.*

**Reply:** The soil auger campaigns were conducted with the purpose to both, collecting soil samples for subsequent laboratory analysis as well as allowing for a WTD measurement. The WTD was measured directly in a 120 cm deep borehole left by the auger, the borehole was open approximately 30 minutes from augering to measurement. We agree that a single direct WTD

measurement is associated with uncertainties due to groundwater fluctuations. However, the soil auger based WTD data is the more or less the prime datasource we have for organic rich peatlands. Wells are often placed in the fringe areas or at deeper depths. We agree, a comparison of soil auger based WTD data and well based WTD data is currently missing.

**Plan for revision:** We will compare auger based WTD data with well based WTD data that are in close vicinity of each other. Furthermore, we will highlight the uncertainties of using temporal snapshots of WTD for building a steady-state WTD model. The analysis will feed into an updated data description as well as discussion.

**Additional analysis & implemented changes:**

In order to investigate the agreement between auger and well data we have compared the closest well for each auger observation and vice versa, the closest auger for each well observation. The results are shown in the figure below. The number of data pairs used for the comparison is stated in the figure at each point. For the shortest distance interval (less than 100 m), the auger observations are 10 cm deeper on average. For pairs with a distance less than 300 m the well observations tend to be 10 cm deeper on average. The spatial correlation vanishes around a distance of 500 m where the difference stabilizes around 40 cm.

Sections 2.2 and 4 have been revised.

[Figure]

*Furthermore, it is unclear how many wells were actually positioned within peat soils (OC > 12%). Are the well measurements representative enough for peatland WTD's, or mainly located within the lowland valleys?*

**Reply:** We agree that information is very relevant and currently not provided in the manuscript. However, we are not very concerned that wells from the fringe areas in the lowlands will a affect

the WTD in peat soils (OC > 12%). First of all, wells were selected based on a 200m buffer analysis as well as a well filter depth constraint of 5 m was applied. Second, the covariates will provide the ML model with information that will distinguish the WTD training data with resect to how representative they are for OC > 12%.

**Plan for revision:** In the data description we will provide an overview of the spatial distribution of wells with respect to OC > 12 %, lowland & river valley and 200 m buffer zone.

**Additional analysis & implemented changes:**

Please find an overview of the number of well and auger observations with respect to the organic content (<6%, 6-12% and >12%) in the table below. As expected, the wells are primarily located in the fringe areas of the peat soils. However, the auger observations are evenly distributed across the OC classes which supports the need for this additional source of WTD observations.

Section 2.2 has been revised.

|        | <6%  | 6-12% | >12% |
|--------|------|-------|------|
| **Wells** | 5235 | 353   | 132  |
| **Auger** | 1835 | 1343  | 1618 |

*The model was assigned to predict a draining effect of ditches by setting draining dummy points, which were found to function successfully within the SHAP analysis (Line 210). However, the authors did not discuss the spatial occurrence of ditches and explain why a ditch WTD of 1.21 (Line 101) would be a valid assumption. Are ditches always dry and thus draining the soils? Or do they also supply water during dry conditions? Is this more or less the same throughout the country?*

**Reply:** First of all, a WTD of 1.21 m was used for the dummy points in ditches as well as for drained forests to provide the model with information on deep WTD under those settings. Given the applied WTD transformation, WTD variability below 1 m depth is very insignificant. A value of 1.21 m corresponds to the dry borehole WTD measurements from the soil auger campaigns. Second, the spatial occurrence was derived from a very detailed national surface water dataset. From that dataset, all small to large streams were removed. For this task we used all streams that are currently represented in the national water resources model (DK-Model). Lastly, yes, it is our assumption that a WTD of 1 m or more right next to ditches is realistic for Danish conditions. In most cases the ditches drain the surrounding land, also in summertime, and ditches in Denmark are generally not used to supply water to fields during dry periods, but entirely to drain water away from fields. We also want to highlight that the applied ML model does not directly simulate whether the ditches drain or supply water from the surrounding fields. In fact, both situations are possible, depending on the training data provided. In case training data indicates a WTD > 1 m in vicinity of ditches, the ditches may be supplying water. Vice versa, shallow WTD training data in vicinity of ditches will likely result in a situation where the ditches drain the surrounding fields.

**Plan for revision:** Since it is not explicitly modelled whether ditches drain or supply water, we cannot conduct a quantitative analysis. However, we will revisit the presentation of ditches in the data section as well as add some details to the discussion.

**Additional analysis & implemented changes:**

No additional analysis has been conducted.

Section 2.2 has been revised.

*Model performance and ecosystem carbon balance*

*A mean error of the trained model of -0.20 m for WTD estimates between 0-0.5 m (Table 3) seems quite high, given the high sensitivity of CO2 emissions predicted for this WTD range (Fig. 7a) and the high percentual area with WTD within this range (Line 257). This seems problematic within further analysis. The authors are open about these high uncertainties (i.e. Line 275) and proceed the analysis on a national scale. The mean error of -0.20 m means that the modelled WTD is more shallow as compared to the observed WTD. Would this mean that national estimates of total emissions are highly underestimated, and that emission reductions are underestimated as well? The MAE of 0.08 m (Line 324, Table 3) is based on validation including dummy points. As these points represent a large portion (1/3) of the dataset the model performance statistic seems biased. Therefore, I would suggest to exclude dummy points by default in Table 3 (and refer to these in text) and report performance statistics including dummy points between brackets instead.*

**Reply:** We totally agree that the bias is substantial given the implications of small uncertainties when applying the Gompertz function for $CO_2$ upscaling. We relate the bias to the fact that the training data are associated with a considerable uncertainty. This uncertainty mainly originates from the fact that we use observations from temporal snapshots to model a steady state mean summer WTD. The bias was calculated as obs minus sim and therefore the bias of -0.2 m in the uppermost depth interval means that the model predicts a WTD that is generally deeper than the observations. This has implications for the GHG emissions. For $CO_2$, the baseline emissions can potentially be overestimated which in turn means the reduction potential may have a tendency to be overestimated.

We have investigated to build a bias correction model on top of the ML model. However, this proofed to be unfeasible since there is not correlation between simulated WTD and bias, which is not surprisingly, because if there was a correlation, the ML model would have picked that up.

Table 3 reports the performance for both all training data and well&auger data only (in brackets). We believe it is fair to report the performance including dummy points, because this is what the ML model trained/optimized against. However, we agree that the performance of well&auger data is the more relevant performance quantification for the given application.

**Plan for revision:** In table 3, the performance based on all data will be stated in brackets instead of the performance based on well&auger auger data. In the text, we will highlight that the latter is the more relevant performance quantification for the given application. Uncertainties of the training data that cause the bias will be further emphasized in the discussion.

**Additional analysis & implemented changes:**

No additional analysis has been conducted.

The presentation of table 3 and the text presenting the table in section 3 has been adjusted as introduced above. Section 4 has also been revised.

*It is exciting that the authors found very similar parameterization for the adopted Gompertz function used in Tiemeyer et al. (2020) (Fig. 7a) (the function to estimate NECB based on WTD). However, the amount of net ecosystem carbon balance (NECB) datapoints in the sensitive WTD range is quite low. As the authors already address, many linear relations have been fitted between WTD and NECB within literature. Except for one observation with a WTD > 1 m, a linear relation would not misfit the data in this research. Such a less steep linear relation would distribute the weight away from the currently very sensitive (and steep) 0- 0.5 m WTD-region and result in carbon balance estimations (and emission reduction estimations) that are less vulnerable to apparently quite substantial errors in WTD estimations(especially for 0-0.5 m WTD). I suggest that the authors conduct two sensitivity analyses: (1) the effect of water table bias on GHG emissions, and (2) the effect of Gompertz function on the GHG emissions by stretching, squeezing and using a linear relation.*

**Reply:** We do not have much to add to this well formulated and spot-on comment. Reviewer #2 also pointed out the need to consider a linear model. We agree that the two suggestions for sensitivity runs will be insightful. Due to the WTD transformation, the model's reliability for WTD greater than 1 m is somewhat limited. This limitation works very fine with the Gompertz model that flattens around 0.4 m, but it may has some implications to the GHG emissions when applying a linear model.

**Plan for revision:** In the revised manuscript we plan to add a new discussion section where we will explore uncertainties and their effect on the overall GHG estimation as well as for the rewetting potential. We will stick to the Gompertz model for presenting the key results. We will 1) explore the possibilities to fit a linear model to the Danish GHG - WTD data and if it is suitable for GHG upscaling and 2) investigate ways to bias correct the GHG sensitive WTD interval.

**Additional analysis & implemented changes:**

We fitted a linear model to the Danish $CO_2$ data. In the figure below, the dotted line is fitted to all data (slope=11.29 intercept=2.75) and the dashed line is fitted to all data except the one WTD observation at ~1.1 m (slope=15.50 intercept=1.57). In the revised manuscript, we only show the model fitted to all data. Either way, carbon accumulation for low WTD is not supported by the Danish flux data when using a linear model. Evans et al. (2021) find an intercept of -1.29. We are not completely sure how the intercept would look like for the German $CO_2$ data, but since Tiemeyer et al. (2020) show a cluster of negative $CO_2$ emissions around WTD of 0.1 m, we would expect that their data also supports a negative intercept. In this way, when applying a linear model fitted to the Danish data, the results should be interpreted with caution. We believe that it is fair to conclude that the current Danish $CO_2$ data does not fully support the development of a linear model.

[Figure]

This is further supported by performance measures of the applied Gompertz and linear models against the observed $CO_2$ flux according to Mayer and Butler (1993) that include the modelling efficiency (MEF) and the mean absolute error relative to the observed mean (rMAE). Further model validation parameters were calculated from regression analysis of observed versus predicted (1:1) data plots, including the significance of the correlation coefficient (***, P < 0.001; **, P < 0.01) and 95% confidence intervals for linear regressions. Based on the results in the figure below we feel confident to conclude that the Gompertz model provides the better representation of the WTD response function with the current knowledge base of Danish $CO_2$ fluxes.

[Figure]

The two sensitivity scenarios requested by the reviewer as well as a third sensitivity scenarios testing a shorter GWP (comment below) are presented in the table below.

In the table below, the column **Presented Results** summarizes the findings presented in the result section, i.e., applying the Gompertz model fitted to the Danish data for GWP100 and mean annual WTD.

The row GHG emission sums the annual emissions from $CO_2$, $CH_4$ and $N_2O$ as $CO_2$ equivalents for OC>12%. The row **Reduction Potential** assumes a full rewetting of the ~74,000 ha with OC>12%.

The remaining three rows reflect the differences between wet and dry prioritization scenarios (see red arrows in updated figure 9 in manuscript).

The **GWP20** scenario applies 20-year global warming potentials instead of 100-year. We used GWP values from IPCC AR6, which means that the GWP for $CH_4$ changes from 27 kg $CO_2$ to 81 kg $CO_2$. For $N_2O$ it is 273 kg $CO_2$ for both GWP20 and GWP100. Increasing the GWP for $CH_4$ results in an increased overall annual GHG emission. The $CH_4$ contribution is highest for shallow WTD, which negatively impacts the reduction potential, which is lower in this scenario. The relative differences between the rewetting scenarios are not really affected by the GWP.

From the cross validation, we concluded a **WTD bias** of -0.2 m (obs minus sim) for the uppermost WTD interval 0-0.5m, indicating that the model simulates the water table too deep. In order to explore the sensitivity of the WTD bias we have defined a scenario where the entire WTD map is corrected with the 20 cm bias, i.e, the WTD map is 20 cm closer to surface across the domain. A WTD closer to the surface results in a lower overall emission. The differences between wet and dry rewetting scenarios are large, because many areas will have a WTD very close to 0 after bias correction and thereby a reduction potential close to 0.

Applying the **linear $CO_2$ model** fitted to all data increases the overall emissions slightly. We can assume that the very shallow and deep WTD intervals have an increased $CO_2$ emission compared to the Gompertz model, whereas the intermediated WTD interval (20-60 cm) has a decreased $CO_2$ emission. Overall, these changes outweigh each other more or less. The reduction potential is much lower compared to Gompertz, which has to be interpreted with care, because of the positive intercept of the linear model. Again, the current Danish $CO_2$ database is likely not ideal to develop a linear model. The differences between wet and dry rewetting scenarios are large. This relates to high $CO_2$ emissions for deep WTD. This relates to limitations of simply extrapolating the linear WTD relationship to WTD > 1m.

| Mg CO2-eq yr-1 | Presented Results | Scenario: GWP20 | Scenario: WTD bias | Scenario: linear $CO_2$ model |
|---|---|---|---|---|
| **GHG emission** | 2.6 | 2.8 | 2.0 | 2.8 |
| **Reduction potential** | 2.0 | 1.4 | 1.4 | 1.1 |
| **Prioritized wet (dry) reduction: 10,000 ha** | 0.1 (0.3) | 0.1 (0.2) | 0.0 (0.3) | 0.0 (0.4) |
| **Prioritized wet (dry) reduction: 20,000 ha** | 0.3 (0.7) | 0.2 (0.5) | 0.0 (0.7) | 0.1 (0.6) |
| **Prioritized wet (dry) reduction: 50,000 ha** | 1.2 (1.6) | 0.9 (1.1) | 0.6 (1.3) | 0.4 (1.0) |

The table above and describing text are added to section 4. The linear model has been included in figure 7 of the revised manuscript. Since the WTD bias is now part of the discussion we omit the WTD sensitivity (summer versus annual WTD map) in figure 9.

*Covariate analysis*

*In the results the authors discuss the impact of land surface temperature (LST)independently of ranked wetness. However, to what extent are these two covariates independent? A crop with deep*

*WTD generally evaporates a higher amount of water, leading to more latent cooling. I would suggest to check the independency of the covariates. Furthermore, the small SHAP values for LST might not justify the high amount of attention in the results section (i.e. Fig. 3 and 5, from Line 228 onwards). In addition, the lower LST in areas of high saturation might not only be caused by more evaporative cooling, but maybe also due to induced heat transport towards deeper soil layers (wet peat soils conduct heat much better).*

**Reply:** We agree, LST and ranked wetness are to some degree related. However, land cover heterogeneity will overrule their relationship. For example, forests have a generally lower LST due to their higher transpiration, but in most cases a low wetness rank. In figure 3, there exists a large, connected patch of low LST in the northern part of the zoom, which shows a low wetness rank, i.e. dry conditions. This patch is forested and here the linkages between LST and ranked wetness are different as over grass or cropland, where we generally see low LST collocated with high wetness ranks, i.e. wet conditions. The ranked wetness is based on the registered crop type by the farmers, which has been analyzed over a period of 5 years to aggregate the data to a single wetness rank. Uncertainties lie in the crop registration as well as in the simplification to derive a ranked wetness. The wetness rank is constant for an entire filed whereas the LST shows variability from cell to cell, which can inform the model of subfield WTD variability, which becomes relevant for larger fields. Based on these arguments we believe that both, LST and wetness rank are of high relevance in the applied ML model.

Yes, the SHAP values for LST may not be the most dominant ones, but based on the top six covariates presented in figure 4, it is among the ones that has a high spatial variability, potentially revealing sub field variability, which should not be forgotten when analyzing the results from the SHAP analysis.

With respect to the evaporative cooling, we interpret the data in the way that a shallow WTD sustains a high water supply for plants for transpiration. We have not considered variations in heat conductance of dry and wet soils.

**Plan for revision:** We will emphasize that evaporative cooling is not the only process that can cause low LST. Moreover, the reasoning of using both, LST and ranked wetness in the model building will be discussed. The importance of both, SHAP values and spatial variability will be addressed in the revised manuscript.

**Additional analysis & implemented changes:**

No additional analysis has been conducted.

Section 3.1 has been revised.

*Total GHG emissions and reduction pathways*

*Given the urgency of climate action, I would suggest to include 25 year global warming potential to calculate total GHG budgets complementary to the 100 year timeframe that is currently applied in Fig. 8 and 9. In this case, the research would address both the short- and long-term effects of rewetting strategies.*

**Reply:** We agree.

**Plan for revision:** The implications of 100- and 25-year global warming potentials will be presented in the results and in the discussion section.

**Additional analysis & implemented changes:**

The GWP20 scenario has been added to the discussion, see comment above.

*Other remarks:*

*- I would suggest aligning the x-axis of Fig. 4 and the legend of Fig. 5 in order to facilitate a constant direction of thinking (i.e. negative SHAP values are related to shallow WTD).*

**Reply:** We agree, the legends can be aligned better.

**Plan for revision:** Legends in figure 5 and figure 6 will be adjusted so that negative SHAP values are on the left and positive SHAP values in the right.

**Additional analysis & implemented changes:**

The colourbar in the legend of Figure 5 has been flipped.

*- Figure 6 gives insight in the general performance of the GBDT models. I suggest to additionally show how well the models perform within the most important and sensitive WTD range of 0- 1.0 m (maybe with grids of 0.1 m or 0.05 m).*

**Reply:** We agree.

**Plan for revision:** The figure will be adjusted in a way that the top panel shows all WTD data and the bottom panel will focus on well&auger observations between 0 – 1.0 m.

**Additional analysis & implemented changes:**

Figure 6 has been revised as follows:

[Figure]

*- It is unfortunate that peat depth could not be taken into account within the analysis (Line 355, 376). Including such data would likely result in a lower prediction of emissions, which could be stated more directly in the text.*

**Reply:** We agree, such a dataset is crucial since peat depth, in combination with the WTD, marks the lower boundary of the unsaturated peat. In Denmark, mapping of peat depth is currently undertaken, and the potential of using the WTD map proposed herein in combination with the peat depth map for an improved national GHG inventory from 2024 and onwards is currently investigated.

**Plan for revision:** The importance om peat depth will be further underlined in the revised discussion.

**Additional analysis & implemented changes:**

Section 4 has been revised.

**"Water table driven greenhouse gas emission estimate guides peatland restoration at national scale" – bg-2023-23**

[Reviewer comments in *italic blue font*; author replies normal font]

*The paper by Koch et al considers the effect of peatland rewetting on GHG emissions in Denmark. It uses a large dataset of water table measurements to test and validate a model for national upscaling. It then looks at prioritising the rewetting of wetter for drier fields to make recommendations. The manuscript is interesting and timely, considering current interests in peatland rewetting for climate mitigation. It is well-written and I enjoyed reading it. I have only minor suggestions. Note however that I have no experience of machine learning so cannot comment at all on that part of the manuscript.*

**Reply:** We would like to thank the reviewer for their thoughtful review and for acknowledging the novelty and relevance of our work. Please find our point by point replies and concrete plans for the revision below.

*Section 2.1. It might be helpful to readers to specific the climate zone somewhere here. A few words of extra text around L82 specifying agriculture (cows + specific kinds of crops?) might also be informative.*

**Reply:** We agree to this point raised by the reviewer. More details on the study site description will surely be appreciated by the readership outside of Denmark.

**Plan for revision:** We will extend the study site description with information on climate and agricultural practices with special emphasize on peatlands.

**Additional analysis & implemented changes:**

Section 2.1 has been revised.

*Section 2.2. Perhaps I misunderstand something, but why is the median used on L89 but mean on L93?*

**Reply:** The median was used for the WTD data originating from wells that had multiple summer observations. The median is preferred here, because it is less sensitive to outliers which we find favorable for curating a WTD training dataset for long-term average summer WTD. The mean was applied to the revisited soil auger sites (2010 and 2021). Here only two measurements are available and calculating a median is not possible, thus the mean was function was applied.

**Plan for revision:** We will revisit the section introducing the WTD and make sure it is stated clearly why the median is applied to the well data but the mean function is applied to the soil auger data.

**Additional analysis & implemented changes:**

Section 2.2 has been revised.

*L160. A slightly pedantic comment, but I'm not sure I'd describe static chamber sampling with syringe sampling and GC analysis (as in Petersen et al 2012) as "state of the art" (there's nothing wrong with this method, but neither is it anything incredibly advanced).*

**Reply:** We agree.

**Plan for revision:** The wording will be changed when introducing the static chamber sampling method.

**Additional analysis & implemented changes:**

Section 2.5 has been revised.

*L171. It is good to see that waterborne C losses are accounted for, using the IPCC defaults.*

**Reply:** Thank your for that acknowledgment.

*L180. 10% seems quite high here. The IPCC Wetlands Supplement has 0.05 (5%) as Fracditch for boreal/temperate drained grasslands and croplands.*

**Reply:** Methane emissions from ditches are only estimated for grids (10 m * 10 m) where we know that ditches are present. For those grids a Fracditch value of 10% corresponds to a 1 m wide ditch water surface, which is generally suitable for Danish conditions. For grids without ditches, Fracditch is set to 0 %. This technicality varies from the IPCC Wetlands Supplement, where the Fracditch is applied to all grids independently whether ditches are present or not, which explains the generally lower value compared to the one we are employing.

**Plan for revision:** The technical implementation on how drains are mapped and how and where the $CH_{4\ ditch}$ is estimated will be extended.

**Additional analysis & implemented changes:**

Section 2.5 has been revised.

*Figure 6. I think measured vs modelled plots are more "honest" when plotted as square panels and the same scale on both axes. That gives a true visual representation of the model.*

**Reply:** We agree that figure 6 can be improved. Reviewer #1 also commented on the figure content and layout. We chose to not plot square panels, because of the applied WTD transformation which causes a strong underestimation for deeper WTD. In order to make the presentation "honest" we decided to plot the 1:1 line. Scales are different for the two panels because the number of points varies substantially.

**Plan for revision:** The figure will be adjusted in a way that the top panel shows all WTD data and the bottom panel will focus on well&auger observations between 0 – 1.0 m. While revisiting the figure we will experiment with square panels and same scales on the colorbars.

**Additional analysis & implemented changes:**

Figure 6 has been revised as follows:

[Figure]

*Figure 7. I wonder how much panel a is skewed by the data point around 1.2 m WTD? Without this point would the relationship be linear, as in Evans et al (2021)? Does it make sense for CO2 emissions to stabilize at 0.4 m WTD? This is mentioned in the discussion but I'm not sure I'm convinced.*

**Reply:** We agree that the choice of function type for building the CO2 response function is crucial. This point was also addressed by Reviewer #1. The literature supports the use of both, Gompertz and linear functions. The stabilization of CO2 at a certain WTD may in reality also relate to a limited peat thickness. Peat depth, in combination with the WTD, marks the lower boundary of the unsaturated peat. Mapping of peat depth is currently undertaken and since the data are not available at present this discussion becomes very hypothetical.

**Plan for revision:** In the revised manuscript we will add a new discussion section where we will explore the choice of function type for building the CO2 response function as a main uncertainty and its effect on the overall GHG estimation as well as for the rewetting potential will be investigated. We will stick to the Gompertz model for presenting the key results. We will explore the possibilities to fit a linear model to the Danish GHG - WTD data and if it is suitable for GHG upscaling. If found suitable, the effect of linear vs Gompertz will be quantified and related uncertainties will be discussed.

**Additional analysis & implemented changes:**

We fitted a linear model to the Danish $CO_2$ data. In the figure below, the dotted line is fitted to all data (slope=11.29 intercept=2.75) and the dashed line is fitted to all data except the one WTD observation at ~1.1 m (slope=15.50 intercept=1.57). In the revised manuscript, we only show the model fitted to all data. Either way, carbon accumulation for low WTD is not supported by the Danish flux data when using a linear model. Evans et al. (2021) find an intercept of -1.29. We are not completely sure how the intercept would look like for the German $CO_2$ data, but since Tiemeyer et al. (2020) show a cluster of negative $CO_2$ emissions around WTD of 0.1 m, we would expect that their data also supports a negative intercept. In this way, when applying a linear model fitted to the

Danish data, the results should be interpreted with caution. We believe that it is fair to conclude that the current Danish $CO_2$ data does not fully support the development of a linear model.

[Figure]

This is further supported by performance measures of the applied Gompertz and linear models against the observed $CO_2$ flux according to Mayer and Butler (1993) that include the modelling efficiency (MEF) and the mean absolute error relative to the observed mean (rMAE). Further model validation parameters were calculated from regression analysis of observed versus predicted (1:1) data plots, including the significance of the correlation coefficient (***, P < 0.001; **, P < 0.01) and 95% confidence intervals for linear regressions. Based on the results in the figure below we feel confident to conclude that the Gompertz model provides the better representation of the WTD response function with the current knowledge base of Danish $CO_2$ fluxes.

[Figure]

In order to investigate the sensitivity with respect to the choice of WTD response function (linear vs Gompertz) we have added a new section to the discussion section where we explore and quantify the sensitivity. We also include 2 additional sensitivity analysis to accommodate the comments raised by the other reviewer. The results are presented below.

In the table below, the column **Presented Results** summarizes the findings presented in the result section, i.e., applying the Gompertz model fitted to the Danish data for GWP100 and mean annual WTD.

The row GHG emission sums the annual emissions from $CO_2$, $CH_4$ and $N_2O$ as $CO_2$ equivalents for OC>12%. The row **Reduction Potential** assumes a full rewetting of the ~74,000 ha with OC>12%. The remaining three rows reflect the differences between wet and dry prioritization scenarios (see red arrows in updated figure 9 in manuscript).

The **GWP20** scenario applies 20-year global warming potentials instead of 100-year. We used GWP values from IPCC AR6, which means that the GWP for $CH_4$ changes from 27 kg $CO_2$ to 81 kg $CO_2$. For $N_2O$ it is 273 kg $CO_2$ for both GWP20 and GWP100. Increasing the GWP for $CH_4$ results in an increased overall annual GHG emission. The $CH_4$ contribution is highest for shallow WTD, which negatively impacts the reduction potential, which is lower in this scenario. The relative differences between the rewetting scenarios are not really affected by the GWP.

From the cross validation, we concluded a **WTD bias** of -0.2 m (obs minus sim) for the uppermost WTD interval 0-0.5m, indicating that the model simulates the water table too deep. In order to explore the sensitivity of the WTD bias we have defined a scenario where the entire WTD map is corrected with the 20 cm bias, i.e, the WTD map is 20 cm closer to surface across the domain. A WTD closer to the surface results in a lower overall emission. The differences between wet and dry rewetting scenarios are large, because many areas will have a WTD very close to 0 after bias correction and thereby a reduction potential close to 0.

Applying the **linear $CO_2$ model** fitted to all data increases the overall emissions slightly. We can assume that the very shallow and deep WTD intervals have an increased $CO_2$ emission compared to the Gompertz model, whereas the intermediated WTD interval (20-60 cm) has a decreased $CO_2$ emission. Overall, these changes outweigh each other more or less. The reduction potential is much lower compared to Gompertz, which has to be interpreted with care, because of the positive intercept of the linear model. Again, the current Danish $CO_2$ database is likely not ideal to develop a linear model. The differences between wet and dry rewetting scenarios are large. This relates to high $CO_2$ emissions for deep WTD. This relates to limitations of simply extrapolating the linear WTD relationship to WTD > 1m.

| Mg CO2-eq yr-1 | Presented Results | Scenario: GWP20 | Scenario: WTD bias | Scenario: linear $CO_2$ model |
|---|---|---|---|---|
| **GHG emission** | 2.6 | 2.8 | 2.0 | 2.8 |
| **Reduction potential** | 2.0 | 1.4 | 1.4 | 1.1 |
| **Prioritized wet (dry) reduction: 10,000 ha** | 0.1 (0.3) | 0.1 (0.2) | 0.0 (0.3) | 0.0 (0.4) |
| **Prioritized wet (dry) reduction: 20,000 ha** | 0.3 (0.7) | 0.2 (0.5) | 0.0 (0.7) | 0.1 (0.6) |
| **Prioritized wet (dry) reduction: 50,000 ha** | 1.2 (1.6) | 0.9 (1.1) | 0.6 (1.3) | 0.4 (1.0) |

The table above and describing text are added to section 4. The linear model has been included in figure 7 of the revised manuscript. Since the WTD bias is now part of the discussion we omit the WTD sensitivity (summer versus annual WTD map) in figure 9.

---

## Author Response (AR2)

**Review round 2 – Publish subject to minor revisions (review by editor)**

**"Water table driven greenhouse gas emission estimate guides peatland restoration at national scale" – bg-2023-23**

[Reviewer/AE comments in *italic blue font*; author replies normal font]

*AE decision:*

*Overall, both reviewers were happy with the changes you made, however, there was still some concern about how you present the uncertainty of your results. Please address the reviewers comments, potentially by adding a range of uncertainty based on your sensitivity analysis.*

*Reviewer 1:*

*The authors have done a thorough job of addressing my comments. The new discussion, and adjusted fig. 7, about linear vs Gompertz fits is particularly welcome.*

*Reviewer 2:*

*The revisions improved the manuscript substantially and most of my concerns are alleviated. I think the main findings (conclusions and abstract) should be presented more carefully, especially on total peatland emissions and emission reductions. Readers may overlook limitations of the study and assume outcomes are extremely solid. Perhaps the authors can implement ranges that are found within the sensitivity analysis. After improving the conclusions I think the manuscript should be accepted.*

**Reply:** We would like to thank the reviewers for acknowledging that our revisions have led to an improvement of our work. We agree that the results of the sensitivity analysis should be mentioned in the conclusions and abstract to disclose uncertainties related to the emission estimate and the estimated reduction potential. To follow the suggestion made by reviewer 2 to summaries the scenario runs, we calculated the coefficient of variation, which we presented in the discussion section, and results are now presented in abstract and conclusion.